# Striding into Clarity: Wearable Sensor-Driven Estimation of Knee Adduction Moment, Unveiling the Black Box with Sequence-Based Neural Networks and Explainable Artificial Intelligence

**Jasmine Y. Liang, Hongyi Bian, Wensheng Zhang, Carl K. Chang, Li-Shan Chou\***

Iowa State University

yupinl@iastate.edu, hobian@iastate.edu, wzhang@iastate.edu, chang@iastate.edu, chou@iastate.edu

## Abstract

Knee adduction moment during walking has been reported as a sensitive biomechanical marker for predicting the risk of knee osteoarthritis. The traditional method of estimating the knee adduction moment relies on the inverse dynamics approach, primarily limited to laboratory settings due to it relies on specialized equipment and technical expertise, which prevents the clinicians' access to the crucial data. Our study employs wearable sensor technology integrated with advanced Artificial Intelligence and Machine Learning algorithms to predict knee moment outcomes with high accuracy. By analyzing attention weight trends, we establish a significant correlation with knee moment dynamics, validating the reliability of our predictive model. This alignment underscores the biomechanical relevance of our approach, offering promising implications for personalized patient care and clinical practice.

## Introduction

Osteoarthritis (OA) is a degenerative joint disease with cartilage loss and decreased joint space (Sellam and Berenbaum 2010). The knee joint is the most reported site for OA, with the incidence being twice of other commonly affected joints (Oliveria, Felson et al. 1995). Patients with knee OA often experience pain, swelling and loss of flexibility during weight-bearing activities, and stiffness after being inactive. Knee OA is one of the highest factors leading to global years lived with disability (YLD) (Vos, Flaxman et al. 2012). The prevalence of knee OA has doubled since the mid-20[th] century, with 37.4 % of adults over 60 years old affected (Wallace, Worthington et al. 2017). Potential risk factors associated with long-term knee symptomatic pain are age, sex, BMI, and history of joint trauma (Zhang and Jordan 2010).

Knee adduction moment (KAM) during walking has been reported as a sensitive biomechanical marker for predicting the risk of knee osteoarthritis (Foroughi, Smith et al. 2009). The knee adduction moment is higher in individuals with severe knee OA and consequently associated with a higher Kellgren-Lawrence (K-L) radiographic grading scale (Sharma, Hurwitz et al. 1998). Patients with onset chronic knee pain were reported with higher mean knee adduction moments when compared to those without symptoms in certain activities, including standing, chair rise, walking, and stair descent (Amin, Luepongsak et al. 2004).

Conventionally, the knee adduction moment is estimated by using the inverse dynamics approach with kinematic data of body movement and ground reactions forces. The whole process significantly restricts testing activities to laboratory settings and its requirement for equipment and technical processing significantly limit clinicians' access to such important data. Recent advances in wearable sensor technologies offer an opportunity to bridge this gap. Moreover, an emerging deep learning model is capable of making predictions based on time series data. Although studies have employed wearable sensors to estimate knee joint moments, time-series information from the input data is not accounted (Stetter, Krafft, et al. 2020).

The primary aim of this study was to estimate knee adduction moments during walking utilizing data acquired from accelerometers. To achieve this, we employed Recurrent Neural Network-Long Short-Term Memory (RNN-LSTM) modeling, a sophisticated approach in the realm of artificial intelligence. Our focus extended beyond prediction by unraveling the RNN-LSTM model's decision-making processes, employing explainable AI techniques. Through this exploration, our study contributed to the biomechanical dynamics of walking and how these advanced models predict and interpret KAM outcomes. The findings of this research hold the potential to inform not only the field of biomechanics but also to advance the broader understanding of explainable AI applications in clinical research.

# Method

## 2.1. Participants

Twelve male and 12 female adults were recruited for the study from the university community. Individuals with any existing pain or musculoskeletal injuries that could potentially impact their walking ability were excluded from the study. Prior to data collection, participants were provided with a detailed explanation of the study objectives and experimental procedures and signed the informed consent approved by the Institutional Review Board.

## 2.2. Procedure

Forty-nine retroreflective markers were placed on specific bony landmarks (Leardini, Sawacha et al. 2007, Leardini, Biagi et al. 2011) to capture the whole-body motion during walking. Marker trajectory data were collected using a 12-camera embedded motion capture system (Qualisys AB, Sweden). Anthropometric reference data were adopted from the initial work of Dempster (Dempster 1955). Concurrently, two tri-axial accelerometers (IMU) were placed at the lateral and medial sides of the femur epicondyles (Trigno Avanti Sensor, Delsys Inc., Natick, MA, USA). To familiarize participants with the experimental protocol and to estimate the individual's average walking speed, three self-selected speed walking trials were conducted prior to formal data collection session. Two photocells separated with three meters were placed to accurately calculate the average walking speed.

$$Avaverage\ self\ selected\ walking\ speed\ (ASSWS)\ (m/s) = \frac{\frac{3\ meters}{time\ measured\ by\ phtotcells}}{3}$$

During all trials, participants wore the same type of shoes to mitigate potential biases resulting from variations in footwear. Marker trajectory and IMU data from each participant were then recorded across three distinct walking speeds, with three trials for each speed. The fast speed was defined as 1.20 times the ASSWS, while the slow speed was set at 0.8 times the ASSWS. This resulted in a total of nine walking trials during the formal data collection session. To maintain consistency in gait patterns for each participant, the walking speeds for all formal trials were carefully calibrated to fall within a range of 95% to 105% of the desired walking speed.

## 2.3. Data Processing and KAM Computing

Marker trajectory data were collected with a sampling rate of 240 Hz and processed using a zero-lag low-pass fourth-order Butterworth filter with a cut-off frequency of 12 Hz (Winter 2009). IMU acceleration data were collected with a sampling rate of 128 Hz, and raw signals were filtered with a 2nd order, zero-lag, and low-pass Butterworth filter with a 12 Hz cut-off frequency (Pitt and Chou 2019). The ground reaction forces are directly measured by two force platforms (AMTI). The knee moment was calculated based on the inverse dynamic theory using Visual3D (C-Motion, Inc., MD).

$$\begin{bmatrix} \tau_1 \\ \tau_2 \end{bmatrix} = \begin{bmatrix} M_{11} & M_{12} \\ M_{21} & M_{22} \end{bmatrix} \begin{bmatrix} \ddot{\theta}_1 \\ \ddot{\theta}_2 \end{bmatrix} + \begin{bmatrix} C_{11} & C_{12} \\ C_{21} & C_{22} \end{bmatrix} + \begin{bmatrix} G_1 \\ G_2 \end{bmatrix}$$

$$\ddot{\theta} = M^{-1}(\theta)\tau + M^{-1}(\theta)C(\theta,\dot{\theta}) - M^{-1}(\theta)G(\theta)$$

$$\ddot{\theta} = M^{-1}(\theta)\tau$$

$$\ddot{\emptyset} = f(\theta,\ddot{\theta})$$

## 2.4. Recurrent Neural Network Model

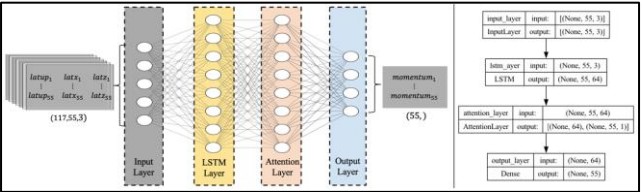

**Figure. 1** depicts an overview of the recurrent neural network (RNN) model structure. Specifically, the model takes as a (55,3) shaped input data, and outputs a regression-based value sequence of shape (55,1). The input data is organized in the format of a time series, encompassing a sequence of 55 temporal intervals. At each interval, the dataset incorporates 3 distinct features, corresponding to the latitude of x, y, and z from a 3-axis accelerometer measurement, respectively. The output is a one-dimensional sequence within the same temporal interval that corresponds to the momentum of the participant's movement.

For this preliminary study, wearable sensor data utilized in the analysis was exclusively from the IMU place at the lateral femur epicondyle. Considering the time-based nature of the input data, we utilized the RNN architecture to enhance the accuracy and effectiveness of our predictive analysis. The constructed model consists of 4 layers, as shown in Figure. 1: Input, LSTM, Attention, and Output. The model was trained using linear acceleration data collected during the stance phase during walking from a sensor positioned on the lateral knee. We choose LSTM as the recurrent layer, given its capability of recognizing and memorizing key steps, therefore enhancing performance in long sequence predictions when compared to a simple RNN layer. In addition to that, we also utilized the attention mechanism in the form of an Attention layer in between the LSTM and Output layer. The attention mechanism provides a means of weighing the relative importance of different time steps in a temporal sequence. This helps to further magnify the model's focus on the most impactful accelerometer readings in regard to predicting the knee moment. Later, we show the extracted attention weights from the Attention layer to explain which time frame during the input sequence made the highest impact in predicting the testing results. The data were randomly assigned to be the training (80%) and testing (20%) datasets.

The rest of the model configurations are as follows:
-The LSTM layer uses "sigmoid" as its activation function.
-The Output layer uses "linear" as its activation function, given this is a regression-based task.
-The model is trained with an Adam optimizer with a learning rate of 1e-3.
-Mean Squared Error (MSE) was used to compute epoch loss.

# Results

## 3.1 Model Prediction Results and Attention Weights

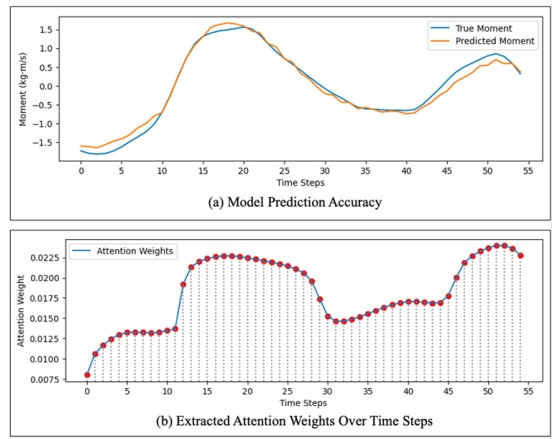

**Figure 2 Model Prediction Results and Attention Weights**
Figure. 2 (a) illustrates a prediction result compared to the ground-truth moment data. The model exhibits a high degree of accuracy (overall MSE = 0.0082) in its predictions, closely mirroring the anticipated measurements throughout the stance phase.

Figure. 2 (b) shows the extracted attention weights during the training phase. The model demonstrates the capability to identify notable shifts in moment within the time series. For example, in the initial phase from time steps 0 to 11, the model intensifies its focus in response to a rising moment. Likewise, in time steps [12,20], [35,43], and [48,52], where the input moment data formed a convex-shaped local minima or maxima, the model's attention was escalated to capture these significant changes in momentum.

This observation further validates that the model not only achieves inference with minimal loss but also does so in a logical, sound and reasonable manner.

## 3.2 Explainable AI: Which features impacted the prediction results?

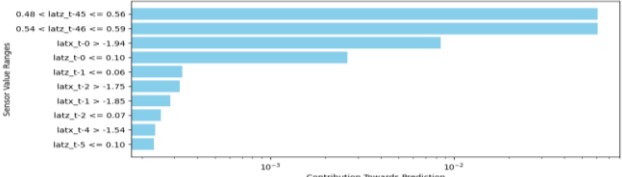

**Figure 3 Top Ten Impactful Sensor Ranges to the Prediction Results**

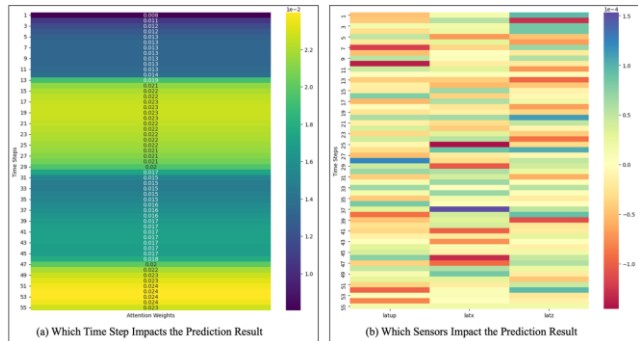

**Figure 4 Global View of the Impactful Sensor Sequences**

In addition to the attention weights for validating the logical soundness of the model, we leveraged an Explainable AI (XAI) tool named "Local Interpretable Model-agnostic Explanations" (LIME) to provide deeper insights into the inferencing outcomes derived from the model's testing phase. It works by perturbating the examined data and observing how these changes affect the model's predictions. Essentially, LIME creates a simpler model that approximates the original model's behavior in local predictions. This simpler model provides insights into the factors employed by the complex model when making its predictions and helps to reveal the reason behind the model's decision for a particular instance.

Figure. 3 shows the top 10 impactful accelerometer readings at a certain time frame. During our testing, we found that the latitude values of the z-axis at time steps 45 and 46, when falling within the ranges of [0.48, 0.56] and [0.54, 0.59] respectively, provide the most significant influence on the model's output. The heatmap in Figure. 4 (a) shows the impact significance of each time frame (disregard each latitude's contribution). In Figure. 4 (b), we present the explanation results by LIME over the training dataset. It provides a clear indication of which specific axis contributed more towards each time step, as well as the over-interference result.

# Discussions

In this study, we employed wearable sensors integrated with advanced AI/ML algorithms to achieve high accuracy

in predicting knee moment outcomes. By leveraging data collected from the IMU sensor, our models were able to analyze intricate patterns and correlations, enabling precise forecasting of knee moment dynamics. This innovative approach not only demonstrates the effectiveness of wearable technology in capturing joint dynamics but also shows potential of AI/ML in enhancing predictive capabilities for knee adduction moment estimation. The robustness and accuracy of our predictions offer prospects for their utilization in analyzing kinetics in clinical settings.

Given the increasing focus on the integration of AI/ML within the domain of digital health applications, it is important to critically evaluate the reliability and trustworthiness of predictions and decisions generated by ML models. As studied in this paper, the attention weight extraction in conjunction with the examination of XAI provides means and trust foundations to the knowledge brought by the ML models. The preliminary findings indicate that the model was able to predict knee adduction moments (KAMs) in a "lightweight" fashion, contrary to the conventional method that requires extensive wearable and environmental sensors. Moreover, it can also be proven that the results came from a reasonable, transparent (human-interpretable) decision-making process as opposed to the pure "black box" mechanism in the traditional sense. Specifically, we observed a consistent alignment between fluctuations in attention weight and corresponding variations in knee moment throughout the duration of the study. This observed coherence suggests that the attention weight assigned by our AI/ML algorithms accurately reflects the biomechanical significance of specific movement patterns or muscle activations, thereby providing a plausible explanation for the observed knee moment outcomes. This enhancement in transparency and interpretability could significantly strengthen the trust in AI/ML-enabled KAM predictions.

Nevertheless, there are still limitations to the current stage of using XAI to provide trust foundations over model predictions/decisions, especially in the medical/health-related context. For instance, standardization remains an area requiring additional exploration. Within the context of KAM prediction, IMU data across three axes were analyzed using an XAI tool to understand the rationale behind prediction outcomes. While it is beneficial to perceive which axis and at what time step brings the most influence on the results at a macro level, the sensor range values of specific decision-making lack clarity regarding their effectiveness. Questions persist about whether a universal threshold can be applied across all KAM scenarios, or if the sensor value ranges should be adjusted based on various factors such as patient characteristics, sensor types, and mounting positions. To gain a deeper understanding of the underlying rationale, further research into these XAI outcomes is necessary.

## Significance

To enhance the transparency and interpretability of our AI/ML models, we provide insights into how predictions are generated based on the underlying rationale. This transparency is essential for building trust and acceptance of AI-driven tools, as it enables users to validate the reliability of predictions and understand the factors driving the model. With an AI/ML model integrated with explainable AI, clinicians can confidently integrate predictive analytics into their practice and further leverage the benefits of wearable sensor technology. Thus, our research not only advances the field of biomechanics but also promotes collaborative between technology and clinical expertise to better improve patient's prognosis.

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
