# OpenReview forum: "Striding into Clarity: Wearable Sensor-Driven Estimation of Knee Adduction Moment, Unveiling the Black Box with Sequence-Based Neural Networks and Explainable Artificial Intelligence "
_AAAI.org/2024/Spring_Symposium_Series/Clinical_FMs — AAAI 2024 SSS on Clinical FMs_

### Official Review · Reviewer_xo9v · 2024-02-22
**A Review of the Innovative Approach to Predicting Knee Moment Dynamics**

**Rating:** 7
**Confidence:** 4

**Review:**

This paper proposes an innovative approach to predicting the dynamic knee moment using wearable sensors, AI, and ML algorithms. The author provides a detailed description of the model structure, training process, and explanatory analysis facilitated by the XAI tool. Here are some review comments:

Methodology: It is strongly recommended that the author explains the physical meanings of each symbol and letter in the formulas on the right side of the second page, as well as the distinctions and connections between the second and third lines of the formulas. Additionally, is the sample size of only 24 participants a bit small?

Results and Discussion: The paper's explanatory analysis of model predictions, particularly using the XAI tool, is very interesting. It is suggested that the author delves deeper into analyzing the model's performance, limitations, and future directions. For instance, a discussion on the model's adaptability to different types of patients or varying environmental conditions would be valuable.

Figures: Figures 3, and 4, along with their respective explanations, are crucial for readers to understand the research results. However, it is advised that the author provides more detailed explanations in the captions and legends of the figures to ensure readers accurately comprehend the information presented in the charts.

Overall, this paper presents a promising study demonstrating how the integration of wearable technology and AI/ML algorithms can predict the dynamic knee moment. Through explanatory analysis and detailed discussions on model performance, the paper has the potential to further strengthen its contributions and practical applications.

---

### Official Review · Reviewer_h7Xr · 2024-02-23
**The paper is interesting but there are no results presented in the paper and furthermore the paper does not include a foundation model or the use of one**

**Rating:** 3
**Confidence:** 3

**Review:**

The paper doesn’t fit the scope of the symposium as it is not a foundation model but rather a supervised learned model.
Clarity: The introduction of the paper is written in clear concise language. When explaining the methods of the paper, however, the language is unclear.
Significance: the work does not appear to be significant within the field of clinical foundation models as it has been trained in what appears to be a supervised manner.
Quality: the quality of the study is subpar with bad figured and poor explanation of methods.
Major points
1.	Figure 1 is basically unreadable; the figure should be bigger.
2.	Figures 3 and 4 are also way too small especially the text. Without a better explanation of time steps they are not very informative.
3.	The sampling frequency specified, but the idea of a timestep is not.
4.	The method is not clearly described.
a.	Explain training objective.
b.	Explain network structure.
5.	Present results
Minor points
6.	In section 2.1 replace 12 with twelwe.
7.	Remove Average self selected walking speed equation 1 to explanation of equation below. Use m (meters) for meters an s (seconds) for time.
Pros
•	New data was aquired for this study.
Cons
•	The study does not include the creation of a foundation model to solve any task. Instead, a supervised model has been developed for assessing the
•	RNN-LSTM seems like a somewhat outdated deep learning model architecture to implement as state of the art.
•	Only 24 subjects were used to train and evaluate a deep learning model, although that is fine for a preliminary study.
•	There are no results presented regarding the performance of the model.
•	It appears that there was no validation dataset, leading me to question how the model was deemed to have trained for long enough.
•	Figures are very bad.

---

### Official Review · Reviewer_ozX1 · 2024-02-23
**Promising results, but needs more clarity in the significance**

**Rating:** 4
**Confidence:** 4

**Review:**

In this work, the authors present a preliminary study for estimating knee adduction moments from data acquired from accelerometers. The authors employ an LSTM-based network that includes an attention layer architecture. Over a cohort of 24 participants, 12 male and 12 female, whole-body motion was captured along with data from two IMU sensors. In addition to evaluating the prediction accuracy provided by the model, the authors analyze the attention weight and an XAI technique, LIME, to gain insights into the prediction made by the model.

This work demonstrates a solid preliminary study into the feasibility of predicting knee adduction moments using IMU data. However, there are several points that could be addressed:

1. There is no report of the overall performance of the model. It seems like the reported example is for a single subject, but there is no report of the evaluation over the entire cohort
2. There is no comparison to other simpler models that would demonstrate the performance improvement. Although this is understandable for a preliminary feasibility study, the overall statistics would be important.
3. The authors report an 80/20 split, but was this split over the participants? Details such as this should be included for a better assessment of the results.
4. Although the authors admit the limitations of the analysis of explainability, how the explainability relates to the trustworthiness of the model is less clear. Moreover,  the approach does not seem particularly innovative, as it applies existing approaches, nor does it provide significant insights into the model and the broader field of biomechanics.

Overall, the results are promising but would recommend another iteration of the manuscript before resubmission.

Additional minor comments:
- ASSWS is not explained in the paper
- The manuscript is over the 2-page limit for the non-traditional track